# Nano-scale architecture of blood-brain barrier tight-junctions

Esther Sasson[1], Shira Anzi[1], Batia Bell[1], Oren Yakovian[2], Meshi Zorsky[3], Urban Deutsch[4], Britta Engelhardt[4], Eilon Sherman[2], Gad Vatine[3], Ron Dzikowski[5], Ayal Ben-Zvi[1]*

[1]Department of Developmental Biology and Cancer Research, Hebrew University of Jerusalem, Jerusalem, Israel; [2]Racah Institute of Physics, Hebrew University of Jerusalem, Jerusalem, Israel; [3]Department of Physiology and Cell Biology, Ben-Gurion University of the Negev, Beer Sheva, Israel; [4]Theodor Kocher Institute, University of Bern, Bern, Switzerland; [5]Department of Microbiology and Molecular Genetics, Hebrew University of Jerusalem, Jerusalem, Israel

**Abstract** Tight junctions (TJs) between blood-brain barrier (BBB) endothelial cells construct a robust physical barrier, whose damage underlies BBB dysfunctions related to several neurodegenerative diseases. What makes these highly specialized BBB-TJs extremely restrictive remains unknown. Here, we use super-resolution microscopy (dSTORM) to uncover new structural and functional properties of BBB TJs. Focusing on three major components, Nano-scale resolution revealed sparse (occludin) vs. clustered (ZO1/claudin-5) molecular architecture. In mouse development, permeable TJs become first restrictive to large molecules, and only later to small molecules, with claudin-5 proteins arrangement compacting during this maturation process. Mechanistically, we reveal that ZO1 clustering is independent of claudin-5 in vivo. In contrast to accepted knowledge, we found that in the developmental context, total levels of claudin-5 inversely correlate with TJ functionality. Our super-resolution studies provide a unique perspective of BBB TJs and open new directions for understanding TJ functionality in biological barriers, ultimately enabling restoration in disease or modulation for drug delivery.

## Editor's evaluation

This study by Sasson and colleagues describes the use of STORM based super-resolution microscopy techniques to examine the structural and functional properties of the blood-brain barrier associated tight junctions. The study reveals some novel findings that could have major importance for our understanding of these tight junction complexes. The role of claudin-5 at the blood-brain barrier is central to the study, and the pattern of expression and localisation of this molecule appears to shed more light on its role.

## Introduction

The blood-brain barrier (BBB) was identified when dye/tracer injected into the blood circulation reached the majority of body tissues but failed to penetrate the brain (*Hagan and Ben-Zvi, 2015*). Endothelial cells (ECs) were identified as the core component of the mammalian BBB when electron microscopy (EM) allowed imaging of fine ultra-structural cell-biology components of BBB cells (*Brightman and Reese, 1969*; *Reese and Karnovsky, 1967*). Horseradish-peroxidase (HRP) used as an EM compatible tracer revealed that BBB ECs lack fenestrations (openings traversing the entire cell width) and exhibit extremely low rates of transcytosis (vesicular transport), both mediating intracellular permeability in

*For correspondence:
ayalb@ekmd.huji.ac.il

Competing interest: The authors declare that no competing interests exist.

peripheral ECs. A central discovery of these studies was that tight junctions (TJs) between neighboring ECs are responsible for intercellular restrictive barrier properties (*Brightman and Reese, 1969*; *Reese and Karnovsky, 1967*). Ever since, BBB TJs became a major focus of the BBB research field; TJs modulation is explored as means to enhance brain drug delivery, and TJs damage is investigated to better understand underlying BBB dysfunctions implicated in diseases (neurodegenerative, neuroinflammatory, trauma etc. [*Bauer et al., 2014*; *Greene et al., 2018*; *Hagan and Ben-Zvi, 2015*; *Kealy et al., 2020*; *Knowland et al., 2014*; *Liebner et al., 2018*; *Sweeney et al., 2018*; *Zhao et al., 2015*; *Zlokovic, 2008*]).

Several gene families encoding integral membrane proteins (e.g. occludin, junctional adhesion molecules [JAMs], claudins, and tricellulins [LSR/Marveld]) and adaptors that link TJs to the cytoskeleton (such as zonula occludens [ZO]), participate in constructing BBB TJs (*Bauer et al., 2014*; *Furuse et al., 1998*; *Furuse et al., 1993*; *Haseloff et al., 2015*; *Knowland et al., 2014*; *Langen et al., 2019*; *Martìn-Padura et al., 1998*; *Morita et al., 1999*; *Nitta et al., 2003*; *Sohet et al., 2015*). Immunofluorescence and imaging with conventional light microscopy provided insights into the molecular components of TJs. Nevertheless, these approaches do not enable proper resolution for nano-scale imaging of TJ architecture. Electron microscopy on the other hand, provides superb resolution to image cellular structures but is much less effective in simultaneously localizing multiple proteins and tracer molecules, limiting our ability to study the molecular architecture of TJs. In order to bridge the gap between these two imaging methodologies and overcome their limitations, we developed an approach to image the BBB with direct stochastic optical reconstruction microscopy (dSTORM [*van de Linde et al., 2011*]) and study TJs at the nano-scale level.

## Results
### Super-resolution microscopy of endothelial tight junctions

We hypothesized that effective imaging of multiple TJ proteins organized in a very tight spatial localization could be achieved with dSTORM. To evaluate this approach we used bEND.3 cells (a mouse brain-derived endothelioma cell-line), cultured in vitro to form a confluent monolayer. These cells were shown to form TJs in a gradual process that includes translocation of claudin-5 from cytoplasmic and general membrane localizations to the cell boundaries in contact between adjacent cells (*Koto et al., 2007*). We confirmed these observations with immunofluorescence labeling claudin-5 and ZO1 in several time points after the cultures reached confluence state (*Figure 1a*). Previous studies also demonstrated that during this process, claudin-5 expression is increased reaching maximal levels by 3 days post-confluence. The monolayer trans-endothelial electrical resistance (TEER), a proxy for TJ function reflecting intercellular restrictive properties, is elevated reaching maximal levels by 7 days post-confluence (*Koto et al., 2007*).

We therefore investigated two states: post-confluence (3–7 days post-confluence) and superconfluence (more than 11 days post-confluence). Using antibodies against claudin-5 and ZO1 for immunofluorescence and dSTORM imaging, we demonstrated a new layer of complexity in TJ organization (*Figure 1b–e*). Most TJ studies use the term 'strands' to describe the organization of TJ proteins in a continuous line around the boundaries of the cells (as seen in *Figure 1a*). Super-resolution imaging allowed us to demonstrate that both claudin-5 (green) and ZO1 (red) are not organized in continuous lines but rather in disrupted lines with discrete clusters, forming bead-like structures (*Figure 1b*). A pronounced change was found in the organization of the two TJ proteins at the cell-cell contacts: at post-confluence, signals along the membrane were more diffused, forming elongated clusters and ZO1 signals intermingled with claudin-5 signals (*Figure 1b*, arrows). In contrast, at superconfluence, claudin-5 signals along the membrane became concentrated in more discrete and shorter foci, flanking ZO1 signals (*Figure 1b*, arrowheads).

Target proteins labeled with antibodies imaged with dSTORM produced resolved signals representing an amplification of actual target numbers (see Materials and methods for details). Resolution of approximately 20 nm allowed us to separate signals and to use these as proxies for the abundance of target molecules, which could be used to compare different states. To quantify the differences in TJ architecture during this in vitro process, we analyzed the images using a custom clustering Matlab code to measure cluster area and number of signals per cluster. These were used to calculate the signal densities for each cluster (see clustering simulation *Figure 1c* and methods for details,

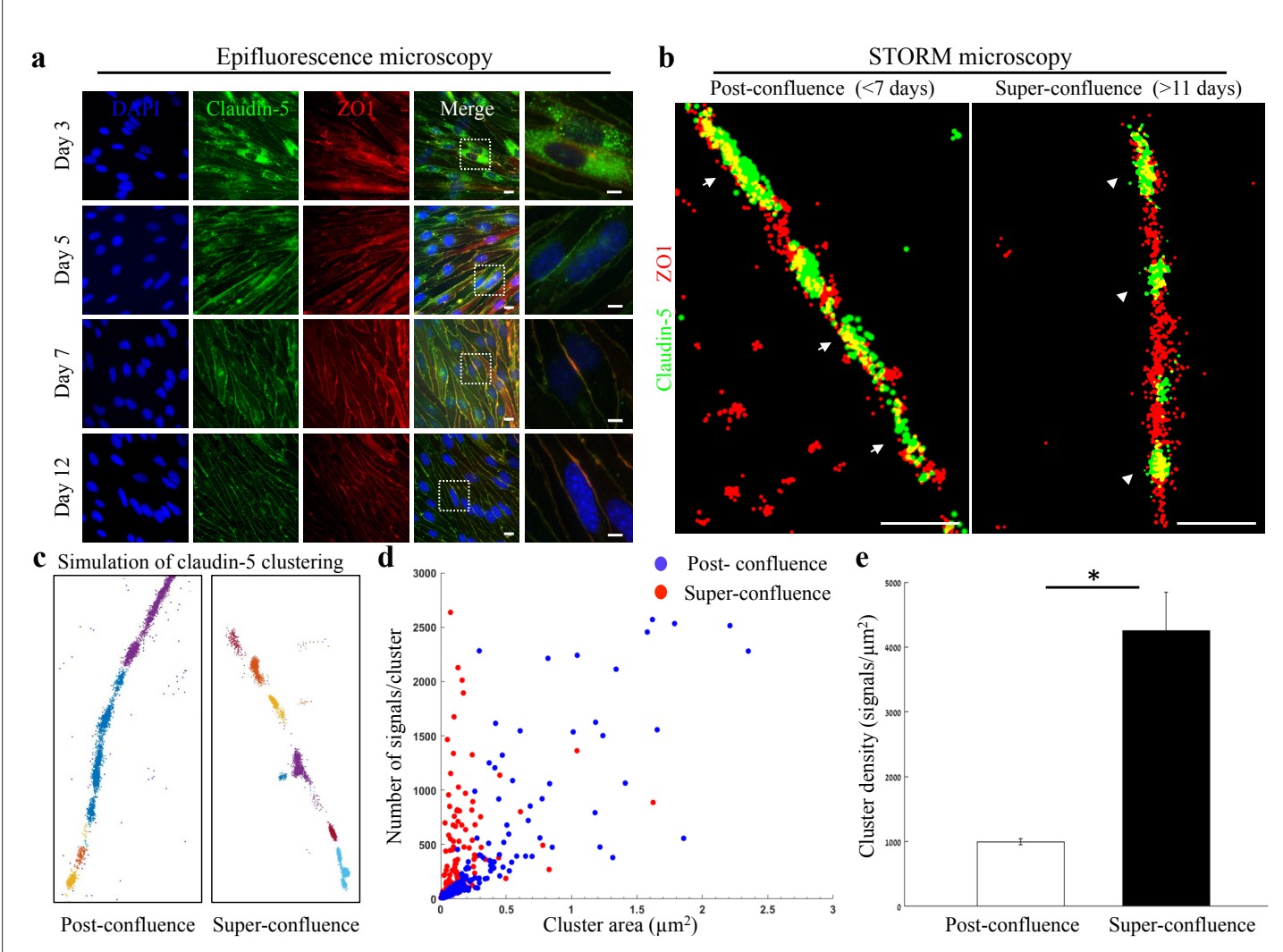

**Figure 1.** Super-resolution microscopy of endothelial tight junctions. In vitro process of TJ maturation is accompanied by TJ architectural changes characterized by the formation of smaller, denser and more discrete clusters of TJ proteins. (**a**) Epi-fluorescent imaging of claudin-5 (green) and ZO1 (red) immunostaining of bEND.3 cells in indicated time points after the cultures reached confluence state. Note translocation of claudin-5 from cytoplasmic localizations into continuous lines around the boundaries of the cells (known as 'strands'), along the in vitro maturation process. Scale bars, 10 μm and 5 μm in insets. (**b**) dSTORM imaging (Gaussian visualization) of claudin-5 (green) and ZO1 (red) immunostaining of bEND.3 cells confluent monolayers. TJ proteins form bead-like structures, especially in the super-confluence state. Claudin-5 signals are more concentrated in discrete and shorter foci (flanking ZO1 signals, arrowheads, right) than in the post-confluence state (arrows, left). During maturation TJ proteins translocate from different cellular locations (left) almost exclusively into the lateral cell membranes (right). Scale bar, 1 μm. (**c**) Examples of dSTORM imaging simulation of claudin-5 in bEND.3 cells used for quantifications of clustering properties (produced by a custom clustering Matlab code, see Materials and methods for details). Signals were defined to be clustered if their 2D location was smaller than 70 nm threshold distance. Cluster pattern visualization showing all points that belong to the same cluster with the same identifying color. (**d**) Quantifications of claudin-5 clustering properties showed a shift towards smaller clusters with more claudin-5 signals per cluster at the super-confluence state. Clusters with higher numbers of signals were more abundant at this late state, especially in clusters with area smaller than 0.3 μm². (**e**) Average claudin-5 cluster density was ~4 fold higher at the super-confluence state than in the post-confluence state (n = 183 clusters (post-confluence) and 281 clusters (super-confluence) in five independent experiments). Data are mean ± s.e.m. *$p < 0.05$ (Two tailed Mann–Whitney U-test).

The online version of this article includes the following source data and figure supplement(s) for figure 1:

**Source data 1.** Related to analyses sumirezed in *Figure 1d, e*.

**Figure supplement 1.** Distance threshold used for quantifications of clustering properties.

**Figure supplement 1—source data 1.** Related to analyses sumirezed in *Figure 1—figure supplement 1b*.

*Figure 1—figure supplement 1*). We could detect a clear shift towards smaller clusters with more claudin-5 signals per cluster at super-confluence (*Figure 1d*). Clusters with higher numbers of signals were more abundant at this late state, especially in clusters with area smaller than 0.3 µm². Average signal density (per cluster) was also higher at the super-confluence state (~4 fold, *Figure 1e*, p < 0.0001).

We further investigated nano-architectural changes in the in vitro context of claudin-5 expression and tight junction function. In line with previous publications (*Koto et al., 2007*), we could confirm that total claudin-5 protein levels in bEND.3 cells rise with time in culture (*Figure 2—figure supplement 1*). Nevertheless, we found that in general, bEND.3 monolayer trans-endothelial electrical resistance (TEER) is relatively low (~40–100 Ωxcm2). Therefore, we turned to an alternative in vitro system that presents substantially superior barrier features with pronounced TJ function and TEER levels closer to those estimated for the in vivo levels: human iPSC differentiation into brain microvascular endothelial-like cells (iBMECs) (*Lippmann et al., 2014*; *Lippmann et al., 2012*; *Vatine et al., 2017*; *Vatine et al., 2019*). In our culturing conditions, TEER levels started at ~500–1000 Ωxcm2 already a day after seeding. We monitored TEER and upon a noticeable elevation of approximately an additional ~1000 Ωxcm2 (2–3 days in culture), we measured claudin-5 protein levels with western blot, permeability in transwells, and in parallel imaged cultures with STORM (*Figure 2*). We noticed that TJ function was improving, with TEER elevation (*Figure 2a*, *Figure 2—figure supplement 2*) and flux decrease (*Figure 2b*), but could not detect noticeable changes in claudin-5 protein levels. STORM imaging revealed that only clustered organization of claudin-5 could be found in these cells (*Figure 2c*), and that pronounced change in nano-scale organization of claudin-5 clusters could be observed; clusters were smaller in area and denser (*Figure 2d–f*) along with improvement in TJ function. The majority of cluster areas were smaller than 0.1 µm², and the average signal density (per cluster) was higher with elevated TEER (~2.3 fold, *Figure 1f*, p < 0.01). In general, clusters of claudin-5 are much denser in iBMECs compared to bEND.3 cells (~8 fold denser), which correlates with differences in TEER.

Altogether, we suggest that the in vitro process of TJ maturation is accompanied by TJ nano-architectural changes characterized by the formation of smaller, denser and more discrete clusters of TJ proteins.

## Molecular organization of mouse cortical BBB TJs

To determine if the organization of TJ components observed in vitro occurs also in the brain, we developed a technique that enabled dSTORM imaging of BBB TJ in brain tissue sections (see Materials and methods for details). First we used fluorescent circulating tracers (*Figure 3b*) and co-staining of claudin-5 together with the endothelial-specific transcription factor ERG (*Figure 3c*) and showed that claudin-5 dSTORM signals are exclusively localized to vascular structures. We found that similar to the in vitro data, claudin-5 exhibits clustered organization also in vivo, demonstrated in both cross and sagittal sections of cortical capillaries (of post-natal mice, *Figure 4b*). The considerably improved resolution of dSTORM imaging could be appreciated when compared to epi-fluorescent images at very high magnifications of the same capillary, under the same microscope settings (*Figures 4b and 5a*).

Embryonic and post-natal development of the BBB provides an opportunity to investigate in vivo TJs maturation; as cortical capillaries acquire their restrictive barrier properties along a gradual developmental process (*Ben-Zvi et al., 2014*; *Butt et al., 1990*; *Daneman et al., 2010*; *Hagan and Ben-Zvi, 2015*; *Langen et al., 2019*; *Saunders et al., 2012*; *Sohet et al., 2015*). From mouse embryonic day 12 (E12) until E15, we and others have previously shown that the newly formed cortical capillaries have not yet acquired their full restrictive barrier properties (*Ben-Zvi et al., 2014*; *Daneman et al., 2010*). We therefore compared claudin-5 organization at E12 and at post-natal day 9 (P9) by analyzing cellular abundance and clustering properties (*Figure 5*, *Figure 5—figure supplement 1*). We assumed that E12 capillaries had a more defused claudin-5 appearance with longer clusters (*Figure 5a*, arrow), but further analysis revealed that these were composed of many small clusters with relatively small gaps between them (example of the two types of clustering simulations, *Figure 5—figure supplement 1*). Claudin-5 clustering properties analysis showed that there were about 2.6 times more discrete clusters per capillary at E12 than at P9 (657 vs 246 clusters in a set of 20 capillaries of each age, *Figure 5b*). Capillary diameter was also significantly larger in E12 than in P9 (*Figure 5a and c*; 11.1 ± 0.47 µm and

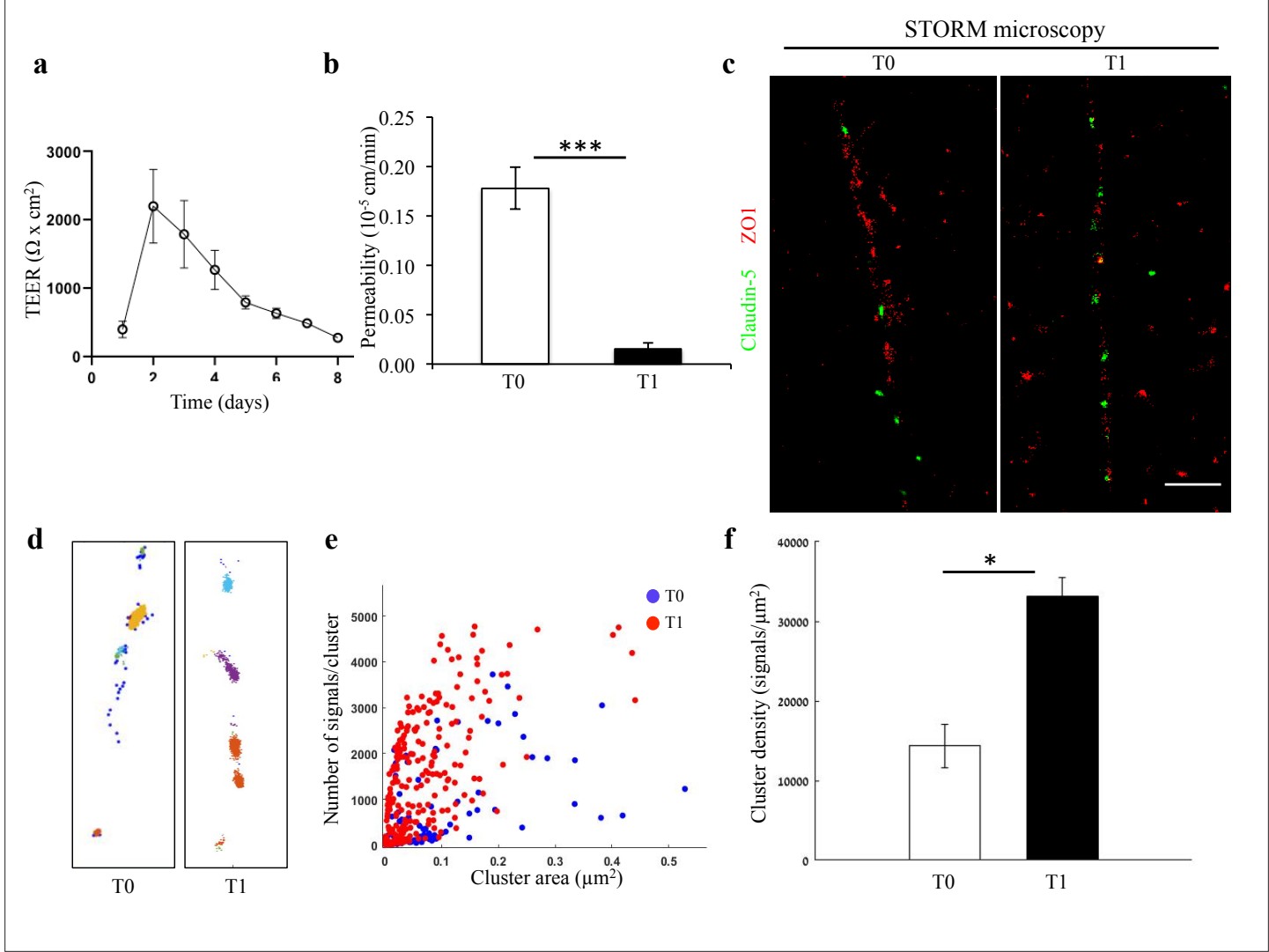

**Figure 2.** Changes in nano-scale architecture correlates with tight junction function. Enhanced TJ function is accompanied by formation of smaller and denser clusters of claudin-5. (**a**) Enhanced TJ function demonstrated by increase in TEER, along the first days of induced human brain microvascular endothelial-like cell (iBMEC) culture (n = 3 experiments/7 inserts. TEER here shows data of a representative experiment. For average change in TEER across all experiments see *Figure 2—figure supplement 2*. n = 4 inserts for permeability). (**b**) Enhanced TJ function demonstrated by reduced permeability to sodium fluorescein (T0 represents low TEER and T1 represents high TEER states, n = 4 inserts). Data are mean ± SD. *** p < 0.0003 (two tailed pair t- test). (**c**) dSTORM imaging (Gaussian visualization) of claudin-5 (green) and ZO1 (red) immunostaining in iBMEC confluent monolayers. Claudin-5 signals are concentrated in discrete and short foci at both time points. Scale bar, 1 μm. (**d**) Examples of dSTORM imaging simulation of claudin-5 in iBMECs used for quantifications of clustering properties (produced by a custom clustering Matlab code, see Materials and methods for details). Cluster pattern visualization showing all points that belong to the same cluster with the same identifying color. (**e**) Quantifications of claudin-5 clustering properties showed a shift towards smaller clusters with more claudin-5 signals per cluster along the improvement in TJ function. (**f**) Average claudin-5 cluster density more than doubled (from 14,341–33,141 signals/μm$^2$) with the improvement in TJ function (n = 90 clusters [in lower TEER] and 278 clusters [in higher TEER] in triplicate cultures of two independent experiments). Data are mean ± s.e.m. *p < 0.01 (two-tailed Mann–Whitney U-test).

The online version of this article includes the following source data and figure supplement(s) for figure 2:

**Source data 1.** Related to analyses sumirezed in *Figure 2a, b, e, f*.

**Figure supplement 1.** Total claudin-5 protein levels in bEND.3 cells are levated along days in culture.

**Figure supplement 1—source data 1.** Raw images of data preesented in *Figure 2—figure supplement 1*.

**Figure supplement 2.** iBMECs TEER is elevated with time in culture.

**Figure supplement 2—source data 1.** Related to analyses sumirezed in *Figure 2—figure supplement 2*.

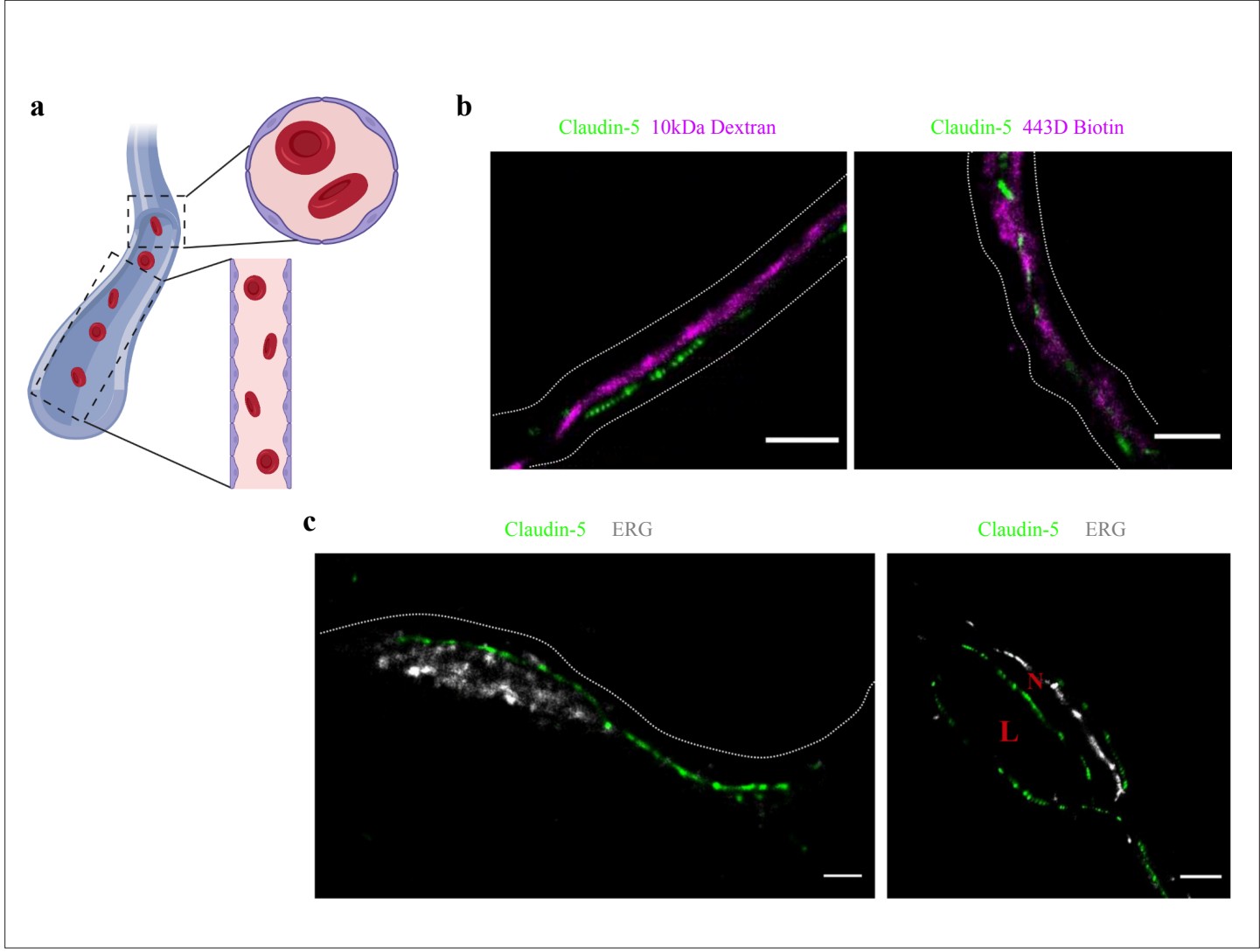

**Figure 3.** Claudin-5 signals in dSTORM imaging are exclusively localized to vascular structures. dSTORM imaging in cortical fixed tissue sections of post-natal day 9 mice. (**a**) Illustration of a vascular structure with cross versus sagittal section directions, and the projected orientation of endothelial cells contact points (Created with https://biorender.com/). (**b**) Claudin-5 staining (green) together with fluorescent circulating tracers (10 kDa dextran, and ~443 Dalton sulfo-NHS-biotin, magenta) are used to demarcate sagittal views of elongated vessels (dashed line, Scale bars, 1 μm). (**c**) Staining for claudin-5 (green) together with the endothelial-specific transcription factor ERG (gray) showing capillary cross and sagittal sections (Scale bars, 1 μm). L – capillary lumen, N – endothelial nucleus. n > 30 capillaries.

5.9 ± 0.39 μm, respectively (mean ± s.e.m), p < 0.0001), which might explain the difference in numbers of clusters per capillary.

Distribution of claudin-5 clusters area in E12 capillaries was skewed towards smaller clusters (with no dramatic difference between the distributions, *Figure 5b*). There was no dramatic difference also in the distribution of signals per cluster (*Figure 5b*) or signal densities between the two groups (*Figure 5—figure supplement 1b*). Thus, there were no obvious changes in claudin-5 clustering properties that correlated with changes in TJs maturation.

In addition to changes in capillary diameter the total number of claudin-5 signals per capillary was significantly higher in E12 than at P9 (14,341 ± 1257 and 3590 ± 372, respectively [mean ± s.e.m], < 0.0001, *Figure 5c*). Normalizing the total number of signals per capillary to its diameter (displayed as 'Normalized cellular Claudin-5', *Figure 5c and d*) resulted in a similar significant difference (*Figure 5d*). Based on this result, we suggest that the total cellular amount of claudin-5 is not a strong predictor of TJ functionality. Our conclusion relates to the developmental and early post-natal BBB (reflected in our data), which might be distinct from the adult BBB.

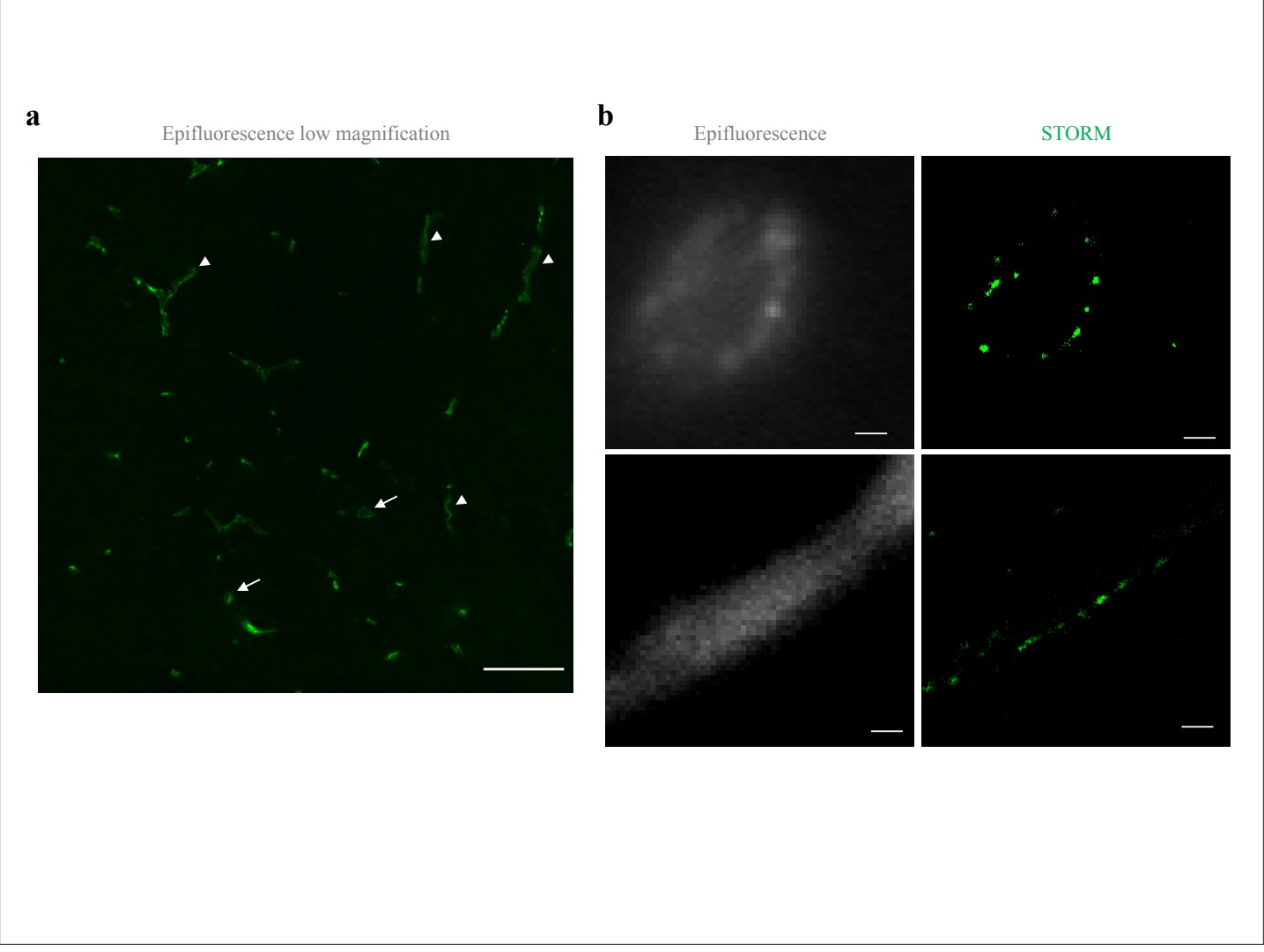

**Figure 4.** Claudin-5 exhibits clustered organization in cortical capillaries. Imaging in cortical fixed tissue sections of post-natal day 9 mice. (**a**) Low-magnification view of claudin-5 staining imaged by epi-fluorescent microscopy showing vascular fragments in cross-sections (arrows) or in sagittal-sections (arrowheads, scale bar, 50 μm). (**b**) Claudin-5 exhibits clustered organization in vivo; dSTORM (right) compared to epi-fluorescent images (left) of claudin-5 immunostaining in P9 cortical capillary cross-section (upper panel) or sagittal- section (lower panel) (scale bars, 1 μm), n > 30 capillaries.

We then expanded the structural and organizational properties examination to include additional TJ proteins in cortical capillaries. Imaging ZO1 and occludin showed that like claudin-5, ZO1 had clustered organization (*Figure 6a* left), whereas occludin was much less organized in discrete clusters and had more dispersed organization patterns (*Figure 6a* right). Based on published biochemical studies, ZO1 is known to physically interact with the C-terminals of both claudin-5 and occludin, which aligns with our imaging data demonstrating signals of all three in close proximity (*Figure 6*, *Figure 6—figure supplement 1*).

While each capillary cross-section presented multiple claudin-5 clusters, we assumed that not all claudin-5 proteins are localized to TJs. Indeed we could detect claudin-5 clusters in close proximity with a lysosomal marker (LAMP1, *Figure 6—figure supplement 2a*), suggesting lysosome localization. In addition, we could detect claudin-5 clusters in close proximity with an ER marker (BiP, *Figure 6—figure supplement 2a*), suggesting ER localization. Therefore we focused on structures where claudin-5 and ZO1 clusters were coupled (*Figure 6b* arrows), reasoning that these might better reflect actual TJs. We analyzed the density of claudin-5 signals in clusters that were coupled with ZO1 clusters and compared it to the density of claudin-5 signals in independent clusters (*Figure 6b*, arrowhead) of both E12 and P9 (*Figure 6b and c*). The average density of P9 claudin-5 clusters that were coupled

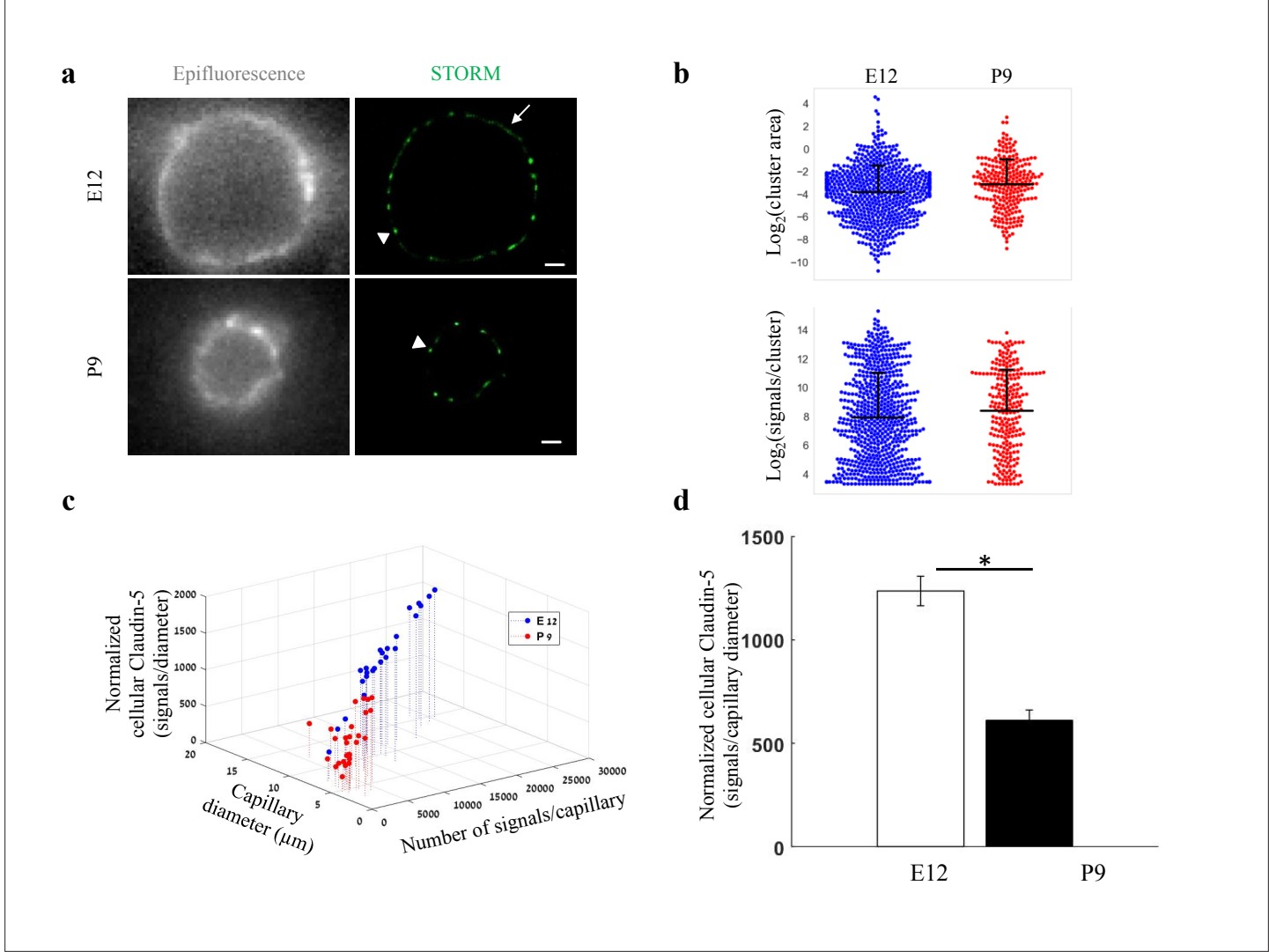

**Figure 5.** Total cellular claudin-5 abundance does not correlate with BBB restrictive properties. Quantifications of claudin-5 levels and clustering properties along developmental BBB maturation (**a**) dSTORM (right) compared to epi-fluorescent images (left) of claudin-5 immunostaining in E12 and P9 cortical capillary cross-sections. Two distinct claudin-5 organizations could be observed; discrete clusters (arrowheads) and a more defused claudin-5 appearance composed of many small clusters with relatively small gaps between them, evident only in E12 capillaries (arrow, see Extended Data *Figure 5—figure supplement 1a* for further analysis). Scale bars, 1 μm. (**b**) Claudin-5 clustering-properties analysis showed that there were about 2.6 times more discrete clusters per capillary at E12 than at P9 (a set of 20 capillaries of each age). Distribution of claudin-5 clusters area in E12 capillaries was skewed toward smaller clusters and the average cluster area was slightly smaller in E12 with no dramatic difference between the distributions (average of 0.309 μm² (E12) vs. 0.3413 μm² (P9)). There was no dramatic difference in the distribution of signals per cluster or signal densities between the two groups (see *Figure 5—figure supplement 1b*). n = 3 pups/embryos, 20 capillaries, 657 clusters (E12) and 246 clusters (P9). Data are mean ± s.e.m. (**c**) Quantifications of *total cellular claudin-5* per capillary crosssection shows a shift towards lower claudin-5 levels and smaller capillary diameter in P9 than in E12 vasculature. Capillary diameter was significantly smaller (5.9 ± 0.39 μm [P9], 11.1 ± 0.47 μm [E12]), total number of claudin-5 signals per capillary was significantly lower (3590 ± 372 (P9), 14,341 ± 1257 [E12]) .(**d**) Normalizing the total number of signals per capillary to its diameter shows the average claudin-5 cellular abundance is significantly lower at P9. n = 25 capillaries (E12) and 27 capillaries (P9) of 3 embryos/pups. Data are mean ± s.e.m. *p < 0.05 (Two tailed Mann–Whitney U-test).

The online version of this article includes the following source data and figure supplement(s) for figure 5:

**Source data 1.** Related to analyses sumirezed in *Figure 5b-d*.

**Figure supplement 1.** Claudin-5 clustering properties.

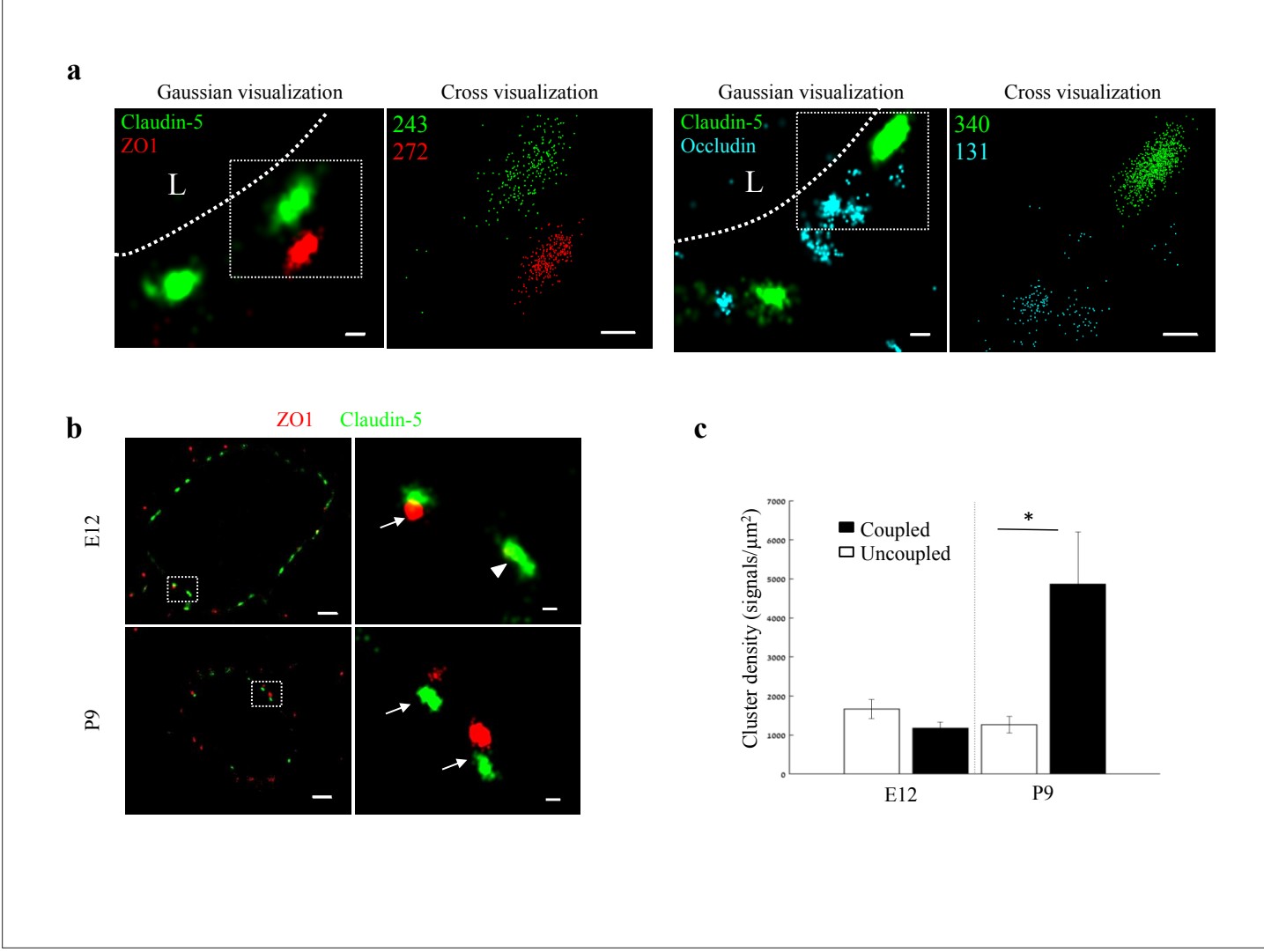

**Figure 6.** Molecular organization of mouse cortical BBB TJs. Nano-scale molecular organization of TJ proteins in cortical capillaries of postnatal wild-type mice (P9). (**a**) Claudin-5 and ZO1 display clustered organization (left) whereas occludin was much less organized in discrete clusters and had more dispersed organization patterns (right). 2D-STORM imaging data demonstrates that signals of all three TJ proteins are in close proximity ('Gaussian visualization' in which signal intensity correlates with localization precision). An inset with magnifications of each cluster (right images) demonstrates the very high-molecular density of TJ proteins ('Cross-visualization' shows all resolved signals where each single-molecule signal displays as a cross). Scale bars, 100 nm. Representative signal numbers are shown. n = 40 capillaries of 4 wild-type pups. L – capillary lumen. (**b**) dSTORM imaging (Gaussian visualization) of claudin-5 (green) and ZO1 (red) immunostaining of E12 and P9 cortical capillary cross-sections. Note that some claudin-5 clusters are coupled with ZO1 clusters (high magnification insets, arrows) while some are independent claudin-5 clusters (arrowhead). Scale bars, 1 μm and 100 nm in insets. (**c**) Average density of P9 claudin-5 clusters that were coupled with ZO1 clusters was ~5 fold higher than independent claudin-5 clusters. The average density of E12 claudin-5 clusters was similar regardless of proximity to ZO1 clusters, and was low compared to P9-independent claudin-5 clusters. n = 40 clusters from 11 capillaries and 43 clusters from 11 capillaries (of three embryos/pups, P9 and E12, respectively). Data are mean ± s.e.m. *p < 0.05 (Two tailed Mann–Whitney U-test).

The online version of this article includes the following source data and figure supplement(s) for figure 6:

**Source data 1.** Related to analyses sumirezed in *Figure 6c*.

**Figure supplement 1.** Molecular organization of ZO1 and occludin in mouse cortical BBB TJs.

**Figure supplement 2.** Claudin-5 clusters are in close proximity to ER and lysosomal markers.

with ZO1 clusters was about five-fold higher (p < 0.0215) than in independent claudin-5 clusters. The average density of E12 claudin-5 clusters was similar regardless of proximity to ZO1 clusters, and was low compared to P9 independent claudin-5 clusters. We concluded that claudin-5 clusters in TJs (based on pairing with ZO1) have higher claudin-5 density in late developmental stages, a structural feature that correlates with BBB maturation.

## ZO1 clustering is independent of claudin-5 in vivo

In order to gain insights on mechanisms underlying Nano-scale molecular architecture of BBB TJs, we imaged claudin-5 null and wild-type littermates cortical capillaries with dSTORM. This approach was intended to enable testing whether claudin-5 being a very abundant transmembrane TJ component might be an organizer of other TJ components. We initially confirmed the specificity of claudin-5

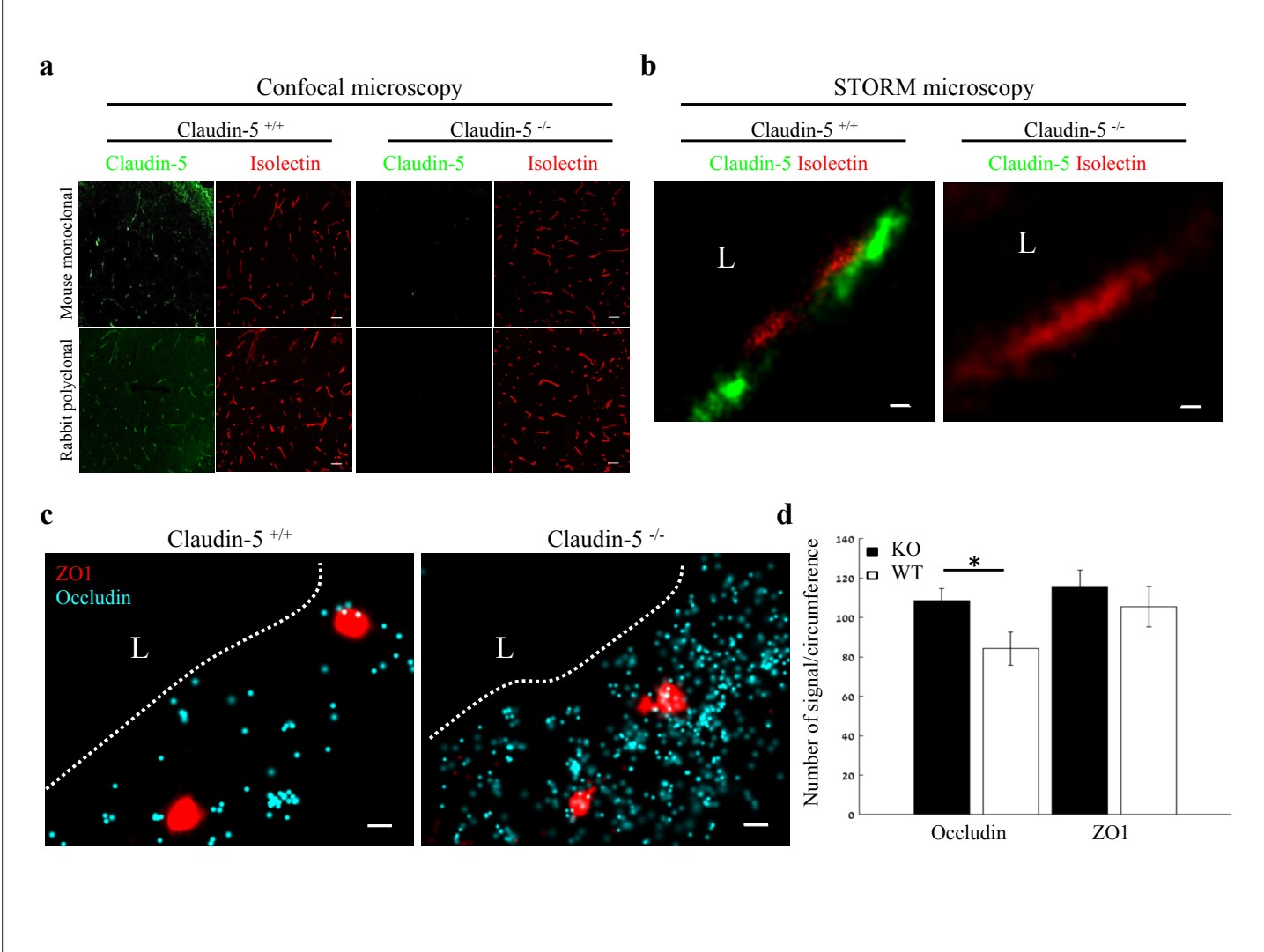

**Figure 7.** Nano-scale organization of both ZO1 and occludin are independent of claudin-5 expression. (**a-b**) Claudin-5 antibodies specificity was confirmed by both confocal microscopy (**a**) Scale bars, 50 μm and dSTORM microscopy (**b**) Scale bars, 0.1 μm, with no detectable staining in E16 cortical null tissues (Isolectin staining used to localize vasculature, n = 4 wild-type, 4 claudin-5 null embryos). (**c**) E16 claudin-5 null and wild-type littermates cortical capillaries imaged with dSTORM display unaltered ZO1 clustering organization and occludin dispersed organization patterns. Scale bars, 100 nm. (**d**) Total cellular signal quantifications revealed that occludin levels were ~1.29 fold higher in claudin-5 null capillaries compared to wild-type. Total cellular ZO1 signal levels were also higher in claudin-5 null capillaries (not statistically significant). Data are mean ± s.e.m. *p < 0.05 (Two tailed Mann–Whitney U-test). L – capillary lumen. n = 48 capillaries of 4 wild-type embryos and 56 capillaries of 4 claudin-5 null embryos.

The online version of this article includes the following source data for figure 7:

**Source data 1.** Related to analyses sumirezed in *Figure 7d*.

antibodies, with no detectable staining in null tissue (confocal microscopy of E16 cortical null and wild-type littermates' tissues, *Figure 7a*). Specificity was also confirmed with dSTORM imaging (*Figure 7b*). Finally, we found that absence of claudin-5 did not altered, ZO1 clustering organization nor it had any effect on occludin dispersed organization patterns (*Figure 7c*). In contrary to the previous conclusions of unaltered molecular composition of claudin-5 null TJs (*Nitta et al., 2003*), quantification of dSTORM imaging revealed that the total cellular occludin levels (normalized to capillary circumference) were higher in claudin-5 null capillaries by ~1.29 fold compared to occludin levels in wild-type capillaries ($P$ < 0.0039, *Figure 7d*). Overall cellular ZO1 expression levels were also higher in claudin-5 null capillaries (not statistically significant). These new findings of molecular alterations in TJ protein levels in the claudin-5 null BBB demonstrate the high sensitivity provided by single-molecule super-resolution imaging, with these molecular changes most probably obscured when tested by other approaches. We concluded that nano-scale organization of both ZO1 and occludin are independent of claudin-5 expression (at least in the embryonic setting). We also believe that these findings warrant a new evaluation of claudin-5 function at BBB junctions.

## Investigating BBB TJ function using super-resolution microscopy

In light of the identified maturation changes in claudin-5 clustering properties, we sought to directly test TJ function in vivo. To this end, we employed tracer challenges and compared TJs permeability at three developmental time points (E12, E16, and P9, *Figure 8*). We performed dSTORM imaging of cortical capillaries following injection of fluorescent tracers to the blood stream and used claudin-5 immunofluorescence to demarcate capillary boundaries and localize TJs in cortical tissues. Similar to the traditional HRP/EM approach (*Reese and Karnovsky, 1967*), dSTORM imaging enables detection of functional BBB TJs with the added value of TJ protein visualization and localization relative to tracer molecules. The HRP/EM approach is not compatible in young embryos and therefore until now, dysfunction of immature TJs was only speculated to underlie capillary hyper-permeability at early stages of BBB development. Thus, we tested E12 TJ function with an in utero embryonic liver tracer injection method that we previously developed to assess BBB permeability during early mouse developmental stages (*Ben-Zvi et al., 2014*). As expected from previous experiments with conventional microscopy (*Ben-Zvi et al., 2014*), with dSTORM imaging we could confirm that E12 capillaries did not restrict movement of tracer molecules across the BBB (*Figure 8a*; 10 kDa dextran, upper left. ~443 Dalton sulfo-NHS-biotin, lower left). Tracer signals could be detected in the basal side or further away, presumably in brain tissue (*Figure 8*, arrowheads). Moreover, we could directly image TJs contribution to this leakage; tracer signals were found intermingled with claudin-5 clusters (*Figure 8*, arrows) and in many cases, we could detect tracer signals in three locations relative to claudin-5 clusters: the luminal side, the cluster area itself and the abluminal side. We interpreted these as direct evidence of tracer leakage across immature TJs. As expected, mature P9 capillaries restrict movement of the vast majority of tracer molecules (*Figure 8*; 10 kDa dextran, upper right. ~443 Dalton sulfo-NHS-biotin, lower right) from the lumen to the basal side (similar results were obtained using Biocytin, data not shown).

In order to examine TJ function along the developmental axis, we also imaged E16 TJs. First, we validated that at E16 cortical TJs were indeed restrictive to the 10 kDa tracers (aligned with general permeability that we previously demonstrated [*Ben-Zvi et al., 2014*]). TJs at this stage were functional and prevented tracer leakage across claudin-5 clusters (*Figure 8a*, middle-upper). Surprisingly, the smaller tracer, sulfo-NHS-biotin, was not restricted to the vessels' lumen, and was evident also on the brain side (*Figure 8a* middle-lower, arrowheads). Therefore at this stage TJs were not as mature as P9 TJs and did not prevent smaller tracer leakage across claudin-5 clusters. These developmental changes in permeability are reflected in quantification of tracer signal density at the abluminal side of the junctions (*Figure 8b and c*). We conclude that BBB TJs might have different maturation time courses for different size selectivity properties and that the dSTORM approach is suitable for in depth investigations of this phenomena in future studies.

## Discussion

We present here a new super-resolution imaging approach for BBB research. Using this approach, we revealed novel structural and functional features of BBB TJs through examination of TJ maturation

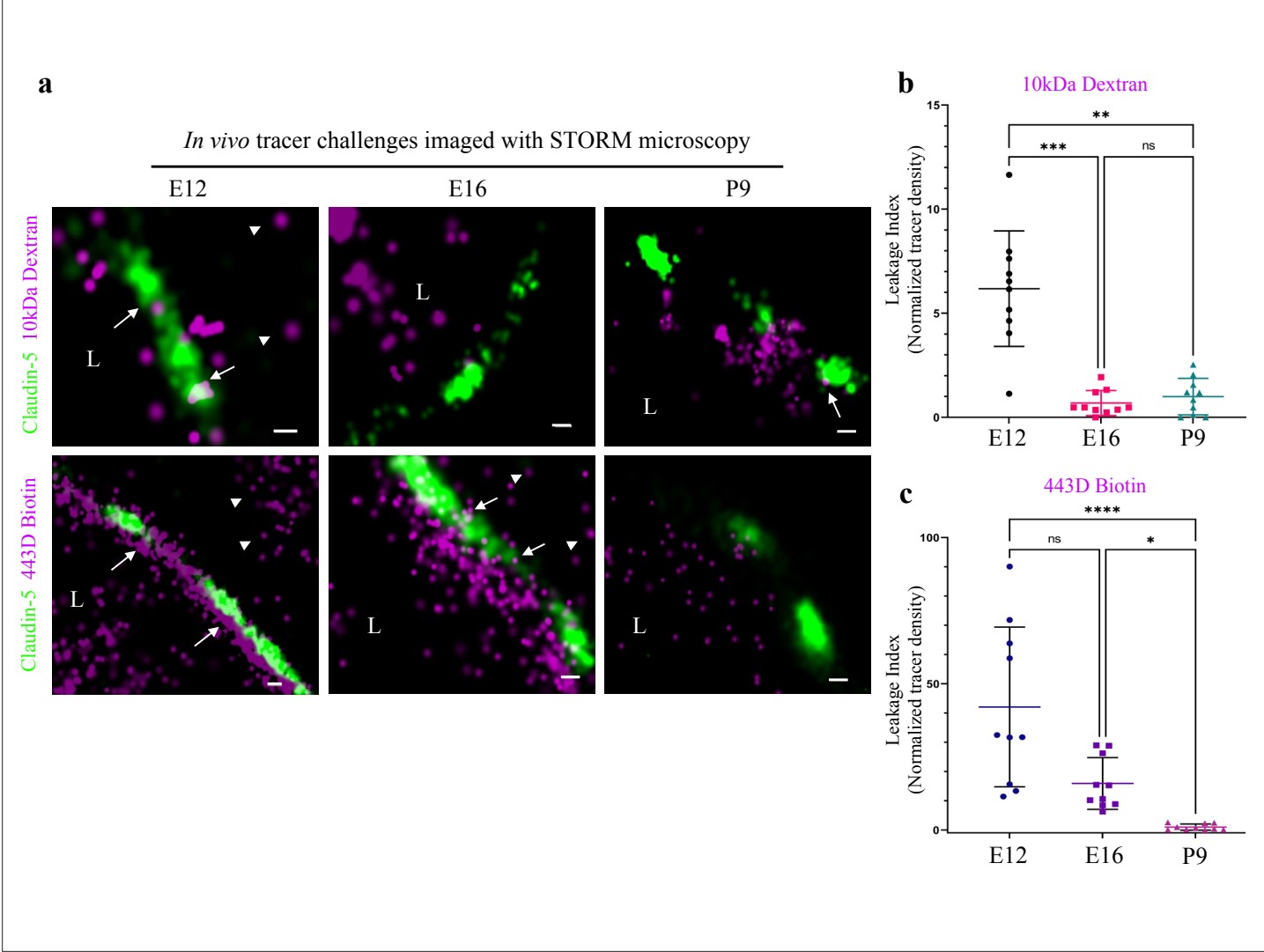

**Figure 8.** Investigating BBB TJ function using super-resolution microscopy. Tracer challenges testing cortical capillaries permeability with dSTORM imaging, provides evidence of leakage across immature TJs. (**a**) E12 TJ function tested with an in utero embryonic liver tracer injection method (***Ben-Zvi et al., 2014***). 10 kDa dextran signals (upper-left) and ~443 Dalton sulfo-NHS-biotin signals (lower-left) were found in the luminal side, intermingling with claudin-5 clusters (arrows) and in the abluminal side and further away (presumably brain tissue, arrowheads). These were interpreted as evidence of tracer leakage across immature TJs. Following trans-cardiac tracer challenges, dSTORM imaging shows P9 capillaries can restrict movement of tracer molecules from the lumen to the brain side (10 kDa dextran upper-right, and ~443 Dalton sulfo-NHS-biotin lower-right). E16 cortical capillaries were previously found to prevent leakage of 10 kDa tracers (***Ben-Zvi et al., 2014***; ***Licht et al., 2015***), also validated here with dSTORM imaging (middle-upper). Surprisingly, at this stage sulfo-NHS-biotin was not restricted to the vessels' lumen, evident also on the brain side (middle-lower arrowheads) and intermingling with claudin-5 clusters (arrows). Scale bars, 100 nm. L – capillary lumen. n = 40 capillaries for each tracer of 3 pups/embryos. (**b–c**), Developmental changes in permeability are reflected in quantification of tracer signal density at the abluminal side of the junctions (10 kDa dextran at b, and 443 Dalton sulfo-NHS-biotin at c). Relative leakage index was calculated as tracer signal density (signals/area) in a fixed area and distance from the abluminal side of the claudin-5 signal, and was normalized to the average signal density at P9 (set as leakage index = 1). n = 10 capillaries for each tracer of 3 pups/embryos, *p < 0.0123, **p < 0.0031,***p < 0.0006, ****p < 0.0001, Kruskal-Wallis test and Dunn's test for multiple comparisons.

The online version of this article includes the following source data for figure 8:

**Source data 1.** Related to analyses sumirezed in ***Figure 8b,c***.

of EC cultures, normal cortical mouse BBB development and claudin-5 null embryos. With super-resolution imaging, we could overcome the limitations of conventional microscopy, which is essential for BBB TJ investigations since it provides proper spatial resolution and enables sensitive quantifications of TJ proteins and tracer molecules.

We provide direct evidence of non-functional TJs in the early embryonic BBB. HRP-EM approaches were not useful for studying early embryonic stages; consequently direct evidence for the cellular pathway mediating leakage in early embryogenesis was missing. A recent mouse retina-vasculature study showed that developmentally, the TJ pathway is already restrictive before the vesicular pathway is blocked (*Chow and Gu, 2017*) and studies in sheep demonstrated similar trends in the brain (*Dziegielewska et al., 1979*). We suggest that in mouse cortical vasculature, the course of barrier-genesis might be different. We found that there are different TJ maturation developmental time-courses for different size selectivities. Thus, even in later stages (E16), when vesicular activity is clearly diminished at the BBB (based on ultra-structural EM studies [*Bauer et al., 1993*]), we found that TJs were not fully mature and mediate leakage of a low-molecular-weight tracer.

We focused on three major TJs components and found sparse occluding vs. clustered ZO1/claudin-5 molecular architecture. Since our current study did not include additional TJ components, we could not differentiate in our analyses between subtypes of TJs and therefore these might represent claudin-5 presumably localized to either bicellular or tricellular junctions.

TJ claudin-5 shifted into denser cluster organization along with in vivo BBB maturation, a finding which is in line with freeze-fracture EM data that suggested cluster organization of EC TJs (*Haseloff et al., 2015*; *Morita et al., 1999*). Therefore, we suggest that confocal imaging description of claudin-5 strands might be a misrepresentation of actual claudin-5 organization. Notably, dSTORM might give rise to some artificial clustering due to antibody staining and fluorophore overcounting (*Burgert et al., 2015*). Hence, our clustering results of claudin-5 and ZO1 should be taken as relative changes in protein organization under the different tested conditions. In addition, our imaging included multiple stains (two different antibodies used for claudin-5 and testing several different fluorophores conjugated to secondary antibody combinations, all resulted in similar clustering data). Moreover, occludin did not show similar clustering properties and finally, we used algorithms to minimize fluorophore overcounting (see Materials and methods [*Ovesný et al., 2014*]).

Previous TJ studies primarily investigated epithelial cells, which have considerably larger volumes than ECs. These studies emphasized a contentious organization of transmembrane proteins along the apical circumference of the cells, providing 'sealing belts'. Our finding of disrupted lines with discrete clusters, forming bead-like structures does not support the concept that claudin-5 fully construct these 'sealing belts'. We can think of three possible explanations for this discrepancy; first, we cannot exclude that our staining approach underrepresents the entire claudin-5 localizations (highly compacted foci might reduce antibodies accessibility). Second, TJ cleft directionality is variable especially in vivo and the 3D organization of the 'sealing belt' might be missed once imaged in 2D. This explanation is incomplete especially once examining the in vitro data where very flat coordinated orientation monolayers are imaged at the cell-glass contacts. Third and most exciting possibility is that other TJ proteins (yet to be discovered) occupy the gaps between claudin-5 clusters; these might be coordinately localized to complete the 'sealing belts'. Such theory aligns with the findings that the claudin-5 null BBB has only partially perturbed sealing properties (*Nitta et al., 2003*).

Our data supports the huge cellular claudin-5 abundance reported by other approaches (e.g. average of ~12,000 transcripts per capillary endothelial cell *Vanlandewijck et al., 2018*, average of ~14,000 [E12] or ~3500 [P9] signals per cross section in our data). Total levels of junctional proteins/transcripts are presented in many studies (measured by western blot, conventional imaging, qPCR/RNAseq etc. [*Armulik et al., 2010*; *Daneman et al., 2010*; *Nitta et al., 2003*; *Vanlandewijck et al., 2018*; *Zhang et al., 2014*]), and down regulation of these components are often used as an indicator for TJ dysfunction (*Alvarez et al., 2011*; *Bell et al., 2010*; *Bell et al., 2012*; *Zhong et al., 2008*). Our findings that claudin-5 expression inversely correlates with BBB tightness in the developmental setting might be distinct from the situation in the adult BBB. Similar to our findings, dSTORM imaging of cultured alveolar epithelial cell TJs, shows increase claudin-5 in response to alcohol exposure together with an increase in paracellular leak (*Schlingmann et al., 2016*). In pathological settings, a minimal threshold of claudin-5 expression might result in TJ leakage. Thus, it will be interesting to test if such a threshold is breached in disease (as suggested for the 22q11 syndrome [*Greene et al., 2018*]).

The differential size selectivity we found in E16 was reminiscent of the seminal study by *Nitta et al., 2003* demonstrating that the BBB of E18.5 claudin-5 null mouse embryos was hyper-permeable to Hoechst (562 Dalton) and to gadolinium (~742 D) but not to 10 KDa dextran or to endogenous albumin (~66 KDa). Following this study, claudin-5 was considered the molecular component of BBB TJs responsible for restricting passage of low molecular weight substances ( < 800 D). Our data indicates that despite the presence of claudin-5, TJs of E16 wild-type mice display similar differential size selectivity as E18.5 claudin-5 null TJs. Accordingly it seems that claudin-5 is not the sole component responsible for restricting passage of low molecular weight substances.

Further investigations are needed in order to evaluate the possible redundancy between different BBB TJ proteins. The increase in occludin expression we describe for claudin-5 null TJ could reflect a developmental compensatory mechanism that might obscure the full contribution of claudin-5 to TJ function. Nitta et al. also suggested that in the absence of claudin-5, claudin-12-based TJs in brain vessels would function as a molecular sieve restricting high-molecular-weight substances but allowing low-molecular-weight substances ( < 800 kDa) to leak into the brain. Recent RNAseq data indicated low expression of claudin-12 in BBB endothelium (approximately 80 fold lower then claudin-5 [*Vanlandewijck et al., 2018*; *Zhang et al., 2014*]), reports farther corroborated by a claudin-12 reporter mouse study indicating expression in many other CNS cell types (*Castro Dias et al., 2019*).

Our findings of unchanged cluster organization of ZO1 in the absence of claudin-5 might be explained by interaction of ZO1 with other TJ proteins (other than occludin which is not clustered) or by recent reports of ZO1 capability of self-organization into membrane-attached compartments via phase separation that can drive TJ formation in epithelia cells (*Beutel et al., 2019*). It also aligns with claudin-5 null embryos having no overt morphological abnormalities (*Nitta et al., 2003*). The other two ZO proteins might also participate in formatting TJ architecture. Based on scRNAseq data, ZO3 is not expressed at the BBB but ZO2 has lower but significant mRNA levels (*Vanlandewijck et al., 2018*). It would be interesting to explore ZO2 nano-scale organization as it shares the same self-organization capacity as ZO1 and therefore might display similar clustering properties (*Beutel et al., 2019*).

Further development of the BBB dSTORM approach will provide additional insights: the ability to image proteins together with membrane lipids and 3D reconstruction of super-resolution imaging are both important focus points for future studies. Deciphering BBB TJ biology with this new approach will aid in evaluating the potential of TJ manipulation for drug delivery and in identifying TJ abnormalities in disease.

## Materials and methods

### Animals

ICR (CD-1, Envigo, Rehovot, Israel) mice were used for embryonic and post-natal BBB functionality assays and dSTORM imaging. Pregnant mice were obtained following overnight mating (day of vaginal plug is defined as embryonic day 0.5). All animals were housed in SPF conditions and treated according to institutional guidelines approved by the Institutional Animal Care and Use Committee (IACUC) at Hebrew University.

The claudin-5 mutant mice (*Nitta et al., 2003*) were kindly provided by Dr. Mikio Furuse (National Institute for Physiological Sciences, Japan). Mice were housed in individually ventilated cages under specific pathogen-free conditions at 22 °C with free access to chow and water. E16 claudin-5 wild-type and null embryos were obtained according to procedures approved by the Veterinary Office of the Canton Bern, Switzerland. Claudin-5 null mutant and wild-type embryos were genotyped using lysates prepared from tips of tails using the following 3 PCR primers: *Cldn5*_UPS: GCCCCTACTAGG ACAGAAACTGGTAG; *Cldn5*_REV1: CAGACCCAGAATTTCCAACGCTGC and PGK-pA-FW1: GCCT GCTCTTTACTGAAGGCTCTT, which provide a 422 bp product for the claudin-5 wild-type allele and a 630 bp product for the claudin-5-knockout allele. PCR cycling conditions were: 4 min 94 °C; 1 min each at 94 °C, 64°C and 72°C; repeated 35 times and a final 5 min elongation step at 72 °C.

### Tissue preparation

After dissection, brains were placed in 4% paraformaldehyde (PFA, Sigma Aldrich) at 4 °C overnight, cryopreserved in 30% sucrose and frozen in TissueTek OCT (Sakura). Frozen brains were cut to 5–8 μm slices for immunofluorescent staining (CM1950, Leica) to produce coronal brain sections.

## Immunohistochemistry

Tissue sections or cell cultures were blocked with 20% goat serum and 20% horse serum, permeabilized with 0.5% Triton X-100, and stained with primary and secondary antibodies (see antibodies table for details). Sample were mounted with freshly made imaging buffer for dSTORM (describe in the dSTORM imaging section) and visualized by dSTORM and epifluorescence, or mounted in Fluoromount G (EMS) and visualized by confocal microscopy. Both a polyclonal and a monoclonal anti-claudin-5 antibody were found to be highly specific in dSTORM, validated with claudin-5 null mice staining, as in confocal imaging (*Figure 7a and b*).

| Epitope | Class | Host | Catalogue number | Company | Dilution |
|---------|-------|------|------------------|---------|----------|
| * Claudin 5 | Monoclonal | Mouse | 35–2500 | Life Technologies | 1:100 |
| ** Claudin 5 | Polyclonal | Rabbit | 34–1600 | Zymed | 1:50 |
| ZO1/TJP1 | Polyclonal | Rabbit | 61–7300 | Thermo Fisher Scientific | 1:200 |
| Occludin | Monoclonal | Mouse | 33–1500 | Thermo Fisher Scientific | 1:50 |
| ERG | Monoclonal | Rabbit monoclonal | ab92513/ EPR3864 | Abcam | 1:200 |
| Lamp1 | Monoclonal | Rat | ID4B | DSHB/ AB_528127 | 1:200 |
| BiP | Monoclonal | Rabbit | C50B12 #3,177 | Cell Signaling Technology | 1:100 |
| GAPDH | Monoclonal | Rabbit | ab181602 | Abcam | 1:400 |

| Fluorophore | Isotype | Catalogue number | Company | Dilution |
|-------------|---------|------------------|---------|----------|
| Alexa fluor647 | Anti-rabbit IgG | 711-605-152 | Jackson | 1:1,000 |
| Alexa fluor647 | Anti-mouse IgG | 711-605-151 | Jackson | 1:1,000 |
| Alexa fluor568 | Anti-rabbit IgG | A11011 | Life Technologies | 1:1,000 |
| Alexa fluor568 | Anti-mouse IgG | A1103-1 | Life Technologies | 1:1,000 |
| Alexa fluor488 | Anti-mouse IgG | 715-545-151 | Jackson | 1:1,000 |
| Streptavidin Alexa fluor647 | Biotin | S32357 | Molecular Probes | 1:800 |
| Alexa fluor647 | Anti-rat IgG | 712-605-153 | Jackson | 1:1,000 |

## Embryonic BBB permeability assay

We used the method we developed and fully described in our previous publication (*Ben-Zvi et al., 2014*). In brief, dams were deeply anesthetized with ketamine-xylazine i.p. (8.5 mg/ml ketamine, 1.5 mg/ml xylazine, in 100 µl saline). Embryos were injected with 5 µl of Dextran, Alexa Fluor647 anionic fixable (D22914, Molecular Probes, 2 mg/ml) or 5 µl of EZ-Link Sulfo-NHS-Biotin (21217, Thermo Fisher Scientific, 1 µg/20 µl), while still attached via the umbilical cord to the mother's blood circulation. Taking advantage of the sinusoidal, fenestrated and highly permeable liver vasculature, dye was injected using a Hamilton syringe into the embryonic liver and was taken up into the circulation in a matter of seconds. After 5 min of circulation, embryonic heads were fixed by 4 hr immersion in 4% PFA at 4 °C, cryopreserved in 30% sucrose and frozen in TissueTek OCT (Sakura).

## Postnatal BBB permeability assay

P9 pups were deeply anaesthetized and 10 µl of Dextran, Alexa Fluor647 anionic fixable (D22914, Molecular Probes, 2 mg/ml) or EZ-Link Sulfo-NHS-Biotin (21217, Thermo Fisher Scientific, 1 µg/20 µl), were injected into the left ventricle with a Hamilton syringe. After 5 min of circulation, brains were dissected and fixed by immersion in 4% PFA at 4 °C overnight, cryopreserved in 30% sucrose and frozen in TissueTek OCT (Sakura).

## In vivo permeability quantifications

Relative leakage index was calculated as tracer signal density (signals/area) in a arbitrary fixed area and distance from the abluminal side of the claudin-5 signal (in ranges of 100–300 nm). For each

tracer and each age, average abluminal signal density of 10 capillaries (from three pups/embryos) was normalized to the average signal density at P9 (set as leakage index = 1).

## Cell culture

The mouse brain endothelioma cell line (bEnd.3) was purchased from American Type Culture Collection (CRL-2299 Manassas, VA, USA) at two occasions, on 2016 and 2019 and was used at 1–2 passages from the original purchased-passage within a year from the purchase. Cells were tested negative for mycoplasma contamination routinely. bEnd.3 cells were cultured with Dulbecco's Modified Eagle's medium high glucose (DMEM), supplemented with 10% fetal bovine serum and 1% penicillin-streptomycin solution (Biological Industries, Beit HaEmek, Israel). Cells were incubated at 37 °C in a humid atmosphere in the presence of 5% $CO_2$. Cells at passages 26–27 were suspended (0.25% Trypsin EDTA B, Biological Industries) and seeded on 24 mm precision coverslips (no. 1.5 H, Marienfeld-superior, Lauda-Königshofen, Germany). Cells were washed with PBS and fixed with 4% PFA (at indicated time point; up to 7 days or more than 11 days post-confluence).

iPSC differentiation to brain microvascular endothelial-like cells (iBMECs) iPSCs from a healthy individual (BGUi012-A) (*Falik et al., 2020*) were cultured between passages 10–17, seeded on Matrigel (Corning) with daily replacement of NutriStem medium (Biological Industries) as previously described (*Falik et al., 2020*, PMID: 32905996). iPSCs were passaged every 6–7 days with Versene (Life Technologies) at a 1:12 ratio. Differentiation into iBMECs was carried out as previously described (*Jagadeesan et al., 2020*; *Vatine et al., 2017*; *Vatine et al., 2019*); cells were passaged and cultured for 2–3 days until reaching a density of $2–3 \times 10^5$ cells/well. Next, medium was replaced with unconditioned medium without bFGF (UM/F: 200 mL of DMEM/F12 [1:1; Gibco], 50 mL knock-out serum replacement [Gibco], 2.5 mL non-essential amino acids [Gibco], 1.25 mL of gluta-max [Gibco], 3.8 uL of β-mercapto-ethanol [Sigma], and 2.5 mL PSA [BI]) and changed daily for 6 days. Medium was then replaced with human endothelial serum-free medium (hESFM, Life Technologies) supplemented with 20 ng/mL bFGF and 10 mM All-trans retinoic acid (RA) (Sigma) (Biomedical Technologies, Inc) for 2 days. Cells were then gently dissociated into single cells with Accutase (StemPro) and plated in hESFM medium at a density of $1 \times 10^6$ cells on transwells (0.4 µm pore size; Corning), coverslips or petri dishes that were pre-coated with a mixture of collagen IV (400 ug/mL; Sigma) and fibronectin (100 ug/mL; Sigma).

## TEER measurements

Trans-endothelial electrical resistance (TEER) was measured every 24 hr following iBMEC seeding. Resistance was recorded using an EVOM ohmmeter with STX2 electrodes (World Precision Instruments). TEER values were presented as Ωxcm2 following the subtraction of an empty transwell and multiplication by 1.12 $cm^2$ to account for the surface area. TEER measurements were measured three independent times for each sample and at least twice for each experimental condition.

## Paracellular permeability measurements

Sodium fluorescein (10 mM) was added to the upper chamber of the Transwells. Aliquots (100 µl) were collected from the bottom chamber every 15 min and replaced with fresh medium. Fluorescence (485 nm excitation and 530 nm emission) was quantified at the end of the experiment with a plate reader. Rate of tracer accumulation was used to calculate Pe values was as previously described (*Vatine et al., 2017*). Monolayer fidelity was confirmed at the beginning and at the end of each experiment by TEER measurements.

iBMECs STORM imaging iBMECs were seeded on 24 mm precision coverslips (no. 1.5 H, Marienfeld-Superior, Lauda-Königshofen, Germany), pre-coated with a mixture of collagen IV (400 µg/mL; Sigma) and fibronectin (100 µg/mL; Sigma). Cultures were fixed in 4% paraformaldehyde for 20 min at room temperature (RT), washed three times with PBS and kept in 4 °C until processing.

## Western blot analysis

Whole cell extracts were isolated using RIPA buffer (50 mM Tris pH 7.4, 150 mM NaCl, 5 mM EDTA pH 8.0, and 1% Nonidet-P40) supplemented with protease inhibitors (Roche). The concentration of the isolated proteins was determined using Bradford reagent (Sigma). A total of 30–50 micrograms of the protein were separated on a 15% polyacrylamide gel and electrophoretically transferred to PVDF

membranes (Millipore). Membranes incubated with the primary antibodies against claudin-5 (1:1000, Zymed 1600–34) or GAPDH (1:400, ab181602, ABCAM) and the appropriate secondary antibodies.

## dSTORM imaging

We used a dSTORM system, which allows imaging at approximately 20 nm resolution by using photo-switchable fluorophores (all dSTORM imaging was done on TIRF mode). Five µm brain slices were mounted on poly-D-lysine coated coverslips (no. 1.5 H, Marienfeld-superior, Lauda-Königshofen, Germany). dSTORM imaging was performed in a freshly prepared imaging buffer containing 50 mM Tris (pH 8.0), 10 mM NaCl and 10% (w/v) glucose with an oxygen-scavenging GLOX solution (0.5 mg/ml glucose oxidase (Sigma-Aldrich)), 40 µg/ml catalase (Sigma-Aldrich), 10 mM cysteamine MEA (Sigma-Aldrich), and 1% β mercaptoethanol (*Barna et al., 2016*; *Dempsey et al., 2011*; *Zhang et al., 2016*). A Nikon Ti-E inverted microscope was used. The N-STORM Nikon system was built on TIRF illumination using a 1.49 NA X100 oil immersion objective and an ANDOR DU-897 camera. 488, 568 and 647 nm laser lines were used for activation with cycle repeat of ~8000 cycles for each channel. Nikon NIS Element software was used for acquisition and analysis; analysis was also performed by ThunderSTORM (NIH ImageJ [*Ovesný et al., 2014*]). Images in 2D were Gaussian fit of each localization; in the N-STORM software.

## dSTORM quantifications

The dSTORM approach we used is based on labeling the target protein with a primary antibody and then using a secondary antibody conjugated to a fluorophore. Thus, resolved signals represent a location that is approximately 40 nm from the actual epitope (assuming the approximation of the two antibodies' length in a linear conformation). The number of signals represents an amplification of the actual target numbers. Amplification corresponds to the primary antibody in the case of a polyclonal antibody (assuming binding to several epitopes in the same protein, which could be reduced by the use of monoclonal antibodies). Amplification also corresponds to several secondary antibodies binding to a single primary antibody and to several fluorophores attached to a single secondary antibody. Nevertheless, resolution of approximately 20 nm allows us to separate signals and to use these as proxies to the abundance of target molecules, which can reliably be used to compare different states.

### Cellular expression level quantifications

We defined the capillary cross-section as an endothelial unit and quantified claudin-5 signals within capillary cross-sections as proxy to total cellular claudin-5 expression levels.

### Cluster area, signal numbers, and signal densities

Single molecule localization microscopy (SMLM) results in point patterns having specific coordinates of individual detected molecules. These coordinates are typically summarized in a 'molecular list' (provided by ThunderSTORM analysis (NIH ImageJ) [*Ovesný et al., 2014*]). In order to define molecular clusters, we analyzed the molecular lists through a custom Matlab code (MathWorks) using the Matlab functions 'Cluster' and 'Linkage', as follows: First, our code calculated distances between each point and all other points in the point pattern of the SMLM image. Then, we set a distance threshold for defining molecules that belong to the same cluster: two points were defined to be clustered if their distance was smaller than the threshold distance (e.g. 70 nm). All points that were clustered with a specific point belong to one cluster (as defined by linkage function). Hence, a point could only be within one cluster. The code then defined and saved the properties of each cluster, such as the area of the cluster, the number of points within the cluster, and the number of clusters. Cluster densities were calculated as number of points divided by each cluster area. Finally, the point patterns were visualized, while showing all points that belong to the same cluster with the same identifying color (*Figures 1c and 2d*, *Figure 1—figure supplement 1*, *Figure 5—figure supplement 1*). The 70 nm threshold distance used for quantifications was determined based on the following parameters: minimal distance could not be below 40 nm (see above antibody labeling strategy); BBB TJs covering continuous contact points, as we evaluated in published TEM imaging data, range approximately up to 100 nm; simulation of claudin-5 density in clusters, measured in different threshold distances between 50 and 100 nm did not yield significant differences (*Figure 1—figure supplement 1*).

Code availability - All custom codes used in this work are freely available at https://github.com/ShermanLab/Cluster-analysis (*Shermanlab, 2021*). These codes use MATLAB 2018b(MathWorks).

## Confocal imaging

Images were captured using Nikon Eclipse Ni confocal microscope, objective X20 with Nikon C2 camera and Nis-Elements software. Images are maximal z-projection of optical sections taken from a 12 µm tissue section imaged with 0.85 µm intervals.

## Epi-fluorescence microscopy

Images presented in *Figures 1a and 4a*, were taken using an Olympus BX51, 10 X/0.3 and 20 X/0.5, with Andor Zyla camera, and Nikon NIS elements software (version D4.5) for both image acquisition and analysis.

## Statistical analysis

All comparisons were performed by two-tailed Mann–Whitney U-tests, or by two tailed pair t- test (as indicated in the figure legends), $p < 0.05$ was considered significant (GraphPad Prism 8.0.1 [244] for Windows, GraphPad Software, San diego, California, USA). For multiple comparisons of leakage index (*Figure 8*), the Kruskal-Wallis test and Dunn's tests for multiple comparisons were used. For the comparison between post confluence and super confluence, we used cluster densities across experiments, for the comparison between ZO1 paired and unpaired claudin-5 clusters we used cluster densities across experiments, for comparisons related to capillaries diameter and total claudin-5 levels we used capillaries across experiments and for comparisons of total occludin levels in wild-type and caludin-5 null embryos we used capillaries across experiments (for exact repetitions see figure legends). Sample size for all immunofluorescence experiments was determined empirically using standards generally employed by the field: a minimum of three animals per group in each experiment, a minimum of four tissue sections of each tissue and a minimum of 10 capillaries per group. In the data set of claudin-5 null and control littermates, the person collecting the data and analyzing was blind to the animal's genotype.

## Acknowledgements

We thank Mss. Sivan Gelb, Kian Atamny and Victoria Miller of the Ben-Zvi group for scientific and writing inputs, Drs. Avihu Klar and Danny Ben-Zvi for scientific inputs, Dr. Gillian Kay for valuable scientific editing, Dr. Norman Grover for his helpful advice regarding statistical analyses, and Ms. Yaara Arad for help with data presentation. We wish to dedicate this work to the memory of Prof. Morris J Karnovsky of Harvard Medical School, a scientific giant and pioneer of the BBB research field. We are grateful for his mentorship and scientific support, and especially for encouraging us to pursue our endeavor of super resolution imaging of BBB TJs.

## Additional information

### Funding

| Funder | Grant reference number | Author |
| --- | --- | --- |
| Israel Science Foundation | 1882/16 | Ayal Ben-Zvi |
| Israel Science Foundation | 2402/16 | Ayal Ben-Zvi |
| Swiss National Science Foundation | 1890809 | Britta Engelhardt |
| Leona M. and Harry B. Helmsley Charitable Trust | 2015PG-ISL007 | Ayal Ben-Zvi |
| Israel Science Foundation | 1621\18 | Gad Vatine |
| Ministry of Science and Technology, Israel | 3-15647 | Gad Vatine |

| Funder | Grant reference number | Author |
|---|---|---|

The funders had no role in study design, data collection and interpretation, or the decision to submit the work for publication.

## Author contributions

Esther Sasson, Conceptualization, Formal analysis, Investigation, Methodology, Validation, Visualization, Writing – original draft; Shira Anzi, Data curation, Investigation, Writing – original draft; Batia Bell, Investigation, Validation; Oren Yakovian, Methodology, Resources, Software, Writing – original draft; Meshi Zorsky, Investigation; Urban Deutsch, Investigation, Resources; Britta Engelhardt, Funding acquisition, Resources, Supervision, Writing - review and editing; Eilon Sherman, Data curation, Resources, Software, Supervision, Writing - review and editing; Gad Vatine, Investigation, Supervision, Writing - review and editing; Ron Dzikowski, Data curation, Investigation, Methodology, Supervision, Writing - review and editing; Ayal Ben-Zvi, Conceptualization, Funding acquisition, Investigation, Supervision, Visualization, Writing – original draft

## Author ORCIDs

Esther Sasson ![ORCID] http://orcid.org/0000-0001-7156-1516
Ayal Ben-Zvi ![ORCID] http://orcid.org/0000-0003-4012-7789

## Ethics

All animals were treated according to institutional guidelines approved by the Institutional Animal Care and Use Committee (IACUC) at Hebrew University (Protocol #MD-15-14449-4).

## Decision letter and Author response

Decision letter https://doi.org/10.7554/eLife.63253.sa1
Author response https://doi.org/10.7554/eLife.63253.sa2

## Additional files

### Supplementary files

• Transparent reporting form

### Data availability

All data generated or analysed during this study are included in the manuscript and supporting files. Tiff and ND2 images and CSV files for STORM imaging are available in deposited archive at EBI (BioStudies accession number S-BSST744). Any additional images of interest or different image formats could be provided upon request to the corresponding author.

The following dataset was generated:

| Author(s) | Year | Dataset title | Dataset URL | Database and Identifier |
|---|---|---|---|---|
| Ayal B, Esther S | 2021 | Nano-scale Architecture of Blood-Brain Barrier Tight-Junctions | https://www.ebi.ac.uk/biostudies/studies/S-BSST744?query=S-BSST744 | BioStudies, S-BSST744 |

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
