## [Editor Report]

This study by Sasson and colleagues describes the use of STORM based super-resolution microscopy techniques to examine the structural and functional properties of the blood-brain barrier associated tight junctions. The study reveals some novel findings that could have major importance for our understanding of these tight junction complexes. The role of claudin-5 at the blood-brain barrier is central to the study, and the pattern of expression and localisation of this molecule appears to shed more light on its role.

---

## [Decision Letter]

**Decision letter after peer review:**

Thank you for submitting your article "Nano-scale Architecture of Blood-Brain Barrier Tight-Junctions" for consideration by *eLife*. Your article has been reviewed by 2 peer reviewers, and the evaluation has been overseen by a Reviewing Editor and Anna Akhmanova as the Senior Editor. The reviewers have opted to remain anonymous.

The reviewers have discussed the reviews with one another and the Reviewing Editor has drafted this decision to help you prepare a revised submission.

Summary:

This study by Sasson and colleagues describes the use of STORM based super-resolution microscopy techniques to examine the structural and functional properties of the BBB associated tight junctions. The study reveals some potentially novel findings that could have major importance for our understanding of the BBB associated TJ complexes. The role of claudin-5 at the BBB is central to the study and the pattern of expression and localisation of this molecule appears to shed more light on its role. The authors propose that the tightening of endothelial junctions during development is correlated with compaction of claudin-5 and ZO-1 into distinct junctional clusters that do not overlap at the matured tight junctions. Overall, this is an important study that will advance the field, although some conclusions are premature and will need more data to substantiate, especially the conclusions related to the reverse correlation of clauidn-5 levels and BBB functional integrity.

Major revisions:

1. Provide in vitro data linking levels of Claudin-5 to tracer flux and TER to better understand relationship to BBB. It's also important to better understand the functionality of the immunoreactive claudin-5 at P9 compared to E12. For example, it is plausible that high levels of claudin-5 developmentally have nothing to do with TJ formation and are redundant for BBB integrity. An in vitro study similar to Figure 1 but examining the levels of claudin-5 and TJ function would be of value here.

2. Reanalyze some current data to provide context (i.e. zoom out to show topology) and controls. For example, it is difficult to orient the tight junctions in panels as no other subcellular information is included. Can the authors provide overview pictures to show the cellular context of the junctions? In addition, since not all junction organizations are the same between cells (for example bicellular or tricellular junctions), how did the authors decide which junctional sites to image for dSTORM? Are some of the immunoreactive spots claudin-5 protein within cell organelles? For example, in cross section, if there is an enrichment of clauidn-5 at a "kissing point", are the other signals coming from claudin-5 being processed and trafficked to the TJ? It is important to show vessel orientation. For example, how does localization look in a sagittal view of an elongated vessel? How does the epifluorescence differ from STORM in this context? For controls, dSTORM imaging yielded a very high (within 20 nm) imaging resolution, so differences in fluorescent signal derived from Claudin-5 and ZO1 may be from the same complex that has changed conformation, leading to different orientation of protein epitopes. The estimated size of primary and secondary antibodies may generate a spacial difference up to 30 nm between the targeted proteins and their tag. In fact the ZO1 signal seems to follow the Claudin-5 condensates with an approximate equal distance in the images in Figures 3a and 3b. Is it possible to image with alternative antibodies that recognize other epitopes on the proteins? Can it be verified that epitope availability is unaffected by the tightening of junctions, to exclude it as explanation for interrupted junction patterns? Is the same localization of Claudin-5/ZO1 observed in superconfluent monolayers if FP-tagged overexpression constructs are taken along as reference?

3. Temper conclusions regarding biology to better fit the presented data, especially with reference to overall BBB function since much of the data is developmental and observational. It's very important to reconcile the results shown with the conclusions being drawn especially in relation to the claudin-5 levels correlating with TJ functionality. The data shown in Figure 2 relate to developmental TJ functionality and this should be clarified and conclusions tempered. For Figure 4, all conclusions need to reference that these data relate to the developmental and early post-natal BBB which will be distinct from the adult BBB.

Revisions Expected in follow up work:

1. Provide in vitro co-localization with other junction proteins to begin to address mechanism. For example, how does the compacted localization of Claudin-5, ZO1 and Occludin relate to the other junctional structures such as the adherens junctions? If ZO-1 does not colocalize with Claudin-5 (or Occludin), then which receptor does it bind to that would explain the clustering?

---

## [Author Response]

Revisions for this paper:Major revisions:1. Provide in vitro data linking levels of Claudin-5 to tracer flux and TER to better understand relationship to BBB. It's also important to better understand the functionality of the immunoreactive claudin-5 at P9 compared to E12. For example, it is plausible that high levels of claudin-5 developmentally have nothing to do with TJ formation and are redundant for BBB integrity. An in vitro study similar to Figure 1 but examining the levels of claudin-5 and TJ function would be of value here.

The design of the in vitro study presented in Figure 1 follows data published by Koto et al., 2007. In that study, Koto et al., already showed that bEND.3 cells have increasing levels of claudin-5 along days in culture in parallel to increasing TJ function (measured by TEER, Koto et al., Figure 1).

As suggested, we replicated this work and measured claudin-5 levels by western blot and confirmed the increasing levels of claudin-5 along days in culture (new Figure2—figure supplement 1). We also attempted to replicate their observed elevation in TEER, but overall under several culturing conditions only modest TEER levels could be measured (~40-100 Ωxcm^2^). Therefore we turned to an alternative in vitro system of substantial superior barrier features with pronounced TJ function and TEER levels closer to these estimated for the in vivo levels:

We used human iPSC differentiation to brain microvascular endothelial-like cells (iBMECs). In our culturing conditions TEER levels starts at around 5001000 Ωxcm^2^ already a day after seeding. We monitored TEER and upon a noticeable elevation of approximately an additional 1000 Ωxcm^2^ (2-3 days in culture), we measured claudin-5 levels (with western blot), measured tracer flux and in parallel imaged cultures with STORM. Only clustered organization of claudin-5 could be found in these cells in both conditions (similar to the in vivo situation). We noticed that in parallel to TJ function improvement (TEER elevation and flux decrease), there was also pronounced change in nanoscale organization of claudin-5 clusters: clusters are smaller in area and denser (new Figure 2). This new data indeed helps to better understand the functionality of the immunoreactive claudin-5 at P9 compared to E12 as suggested by the reviewers:

“it is plausible that high levels of claudin-5 developmentally have nothing to do with TJ formation and are redundant for BBB integrity”.

We also suggested a similar conclusion:

“total cellular amount of claudin-5 is not a strong predictor of TJ functionality”.

We are aware of recent published work regarding the iBMECs having epithelial properties in addition to their high barrier function and endothelial properties (Lu TM et al., PNAS 2021). Therefore we adopted the terminology of ‘endothelial-like cells’. We think that these cells, which express human claudin-5 together with unprecedented in vitro TJ function, serve our purpose of investigating the nano-scale architecture of barrier TJs together with TJ function.

2. Reanalyze some current data to provide context (i.e. zoom out to show topology) and controls. For example, it is difficult to orient the tight junctions in panels as no other subcellular information is included.

The pattern of capillaries and TJ orientation are now showed by localization of claudin-5 signals together with the endothelial specific nuclear staining of ERG (new Figure 3).

Can the authors provide overview pictures to show the cellular context of the junctions? In addition, since not all junction organizations are the same between cells (for example bicellular or tricellular junctions), how did the authors decide which junctional sites to image for dSTORM?

We attempted to use anti-LSR antibodies to investigate the differences in claudin-5 organization between bicellular or tricellular junctions (see Author response image 1). In our opinion, STORM images of LSR staining provided clear locations of tricellular junctions as expected from previous publications showing a clear ‘spot’ (Sohet et al., JCB 2015) only in rare occasions (see Author response image 1), but in general did not allow a straight forward identification of tricellular junctions. Therefore, we now clearly state the limitation of our observations and that all data and analyses might include claudin-5 presumably localized to either bicellular or tricellular junctions (lines 289-292).

**Author response image 1. sa2fig1:** Claudin-5 signals in close proximity to the tricellular marker LSR. Cortical capillary cross-sections of P9 mice presented multiple claudin-5 clusters, of which the minority was in close proximity with the tricellular marker LSR. The majority of LSR signals were diffused and only rare images included clusters of LSR adjacent to what appears as a ‘T shaped’ claudin-5 distribution. Scale bars, 1 µm and 0.1 µm in inset.

Are some of the immunoreactive spots claudin-5 protein within cell organelles? For example, in cross section, if there is an enrichment of clauidn-5 at a "kissing point", are the other signals coming from claudin-5 being processed and trafficked to the TJ?

Indeed our previous analyses were based on such an assumption “While each capillary cross-section presented multiple claudin-5 clusters, we assumed that not all claudin-5 proteins are localized to TJs. Therefore we focused on structures where claudin-5 and ZO1 clusters were coupled (Figure 6b arrows), reasoning that these might better reflect actual TJs”. We now provide additional support with co-staining with an ER marker (BiP) and a lysosomal marker (Lamp1) in STORM imaging, showing close proximity of claudin-5 signals with these markers (new Figure6—figure supplement 2).

It is important to show vessel orientation. For example, how does localization look in a sagittal view of an elongated vessel? How does the epifluorescence differ from STORM in this context?

The pattern of capillaries was identified by localization of claudin-5 signals together with tracers to help demarcate capillaries in sagittal views of elongated vessels (new Figures 3-4) in addition to cross sections (Figures 35). Both STORM and epifluorescence sagittal views of claudin-5 staining are shown.

For controls, dSTORM imaging yielded a very high (within 20 nm) imaging resolution, so differences in fluorescent signal derived from Claudin-5 and ZO1 may be from the same complex that has changed conformation, leading to different orientation of protein epitopes. The estimated size of primary and secondary antibodies may generate a spacial difference up to 30 nm between the targeted proteins and their tag. In fact the ZO1 signal seems to follow the Claudin-5 condensates with an approximate equal distance in the images in Figures 3a and 3b. Is it possible to image with alternative antibodies that recognize other epitopes on the proteins?

We made substantial efforts to find additional antibodies that recognize other epitopes, especially extracellular epitopes. Unfortunately, all the antibodies that we could find are targeted against the intracellular c-termini of the proteins.

With no available reagents, I am afraid we could not provide a clear answer. We are currently working on alternative approaches to label extracellular epitopes, but these will take at least another year to develop.

Can it be verified that epitope availability is unaffected by the tightening of junctions, to exclude it as explanation for interrupted junction patterns? Is the same localization of Claudin-5/ZO1 observed in superconfluent monolayers if FP-tagged overexpression constructs are taken along as reference?

As suggested, we used over expression of eYFP-fused claudin-5 in bEND.3 cells. eYFP was imaged directly with STROM in Super-confluent monolayers (several suitable constructs were kindly provided by Dr. Jörg Piontek [at Charité] and by Dr. Michael Koval [at Emory]).

Clustered localization of claudin-5 similar to that of claudin-5 antibody staining could be observed (see Author response image 2). Nevertheless, once evaluated blindly, half of the samples were found to exhibit interrupted junction patterns and the other half exhibited more continuous patterns. This dichotomy was not observed with antibody staining of endogenous claudin-5, in which the majority of samples were found to exhibit interrupted junction patterns.

**Author response image 2. sa2fig2:** Direct STORM imaging of EYFP-fused claudin-5, overexpressed in bEND. 3 cells. bEND.3 cells were infected with a lenti-viral vector to stably express ctermini EYFP-tagged human claudin-5. Super-confluent monolayers were imaged with STORM to localize the EYFP signals. About half of the samples were found to exhibit interrupted junction patterns (e.g right) and the other half exhibited more continuous patterns (e.g. left). No significant difference in signal density was found between the two patterns. Scale bar, 100 mm.

In order to evaluate whether epitope availability is affected by the tightening of junctions, we analyzed claudin-5 densities in the two patterns of the eYFP over expression samples. Densities distribution was very variable and we could not find significant difference in signal density between the two patterns. Therefore we could not conclude that failing to identify continuous junction patterns in super-confluent monolayers by antibody staining is due to limited epitope availability.

The differences in patterns observed between the eYFP and the endogenous claudin-5 might be due to over expressing the construct. Average claudin-5 density per cluster was considerably higher when imaged with eYFP (~15,000 signals/um^2^) compared to imaging with antibody staining of the endogenous claudin-5 (~4,000 signals/um^2^), but as mentioned, did not correlated with a certain junctional pattern. We could not exclude the possibility that higher eYFP density does not stem from over expression, nor from potential differences between the endogenous mouse claudin-5 gene and the over expressed human claudin-5 gene that we used. In this context, it should be noted that *antibody staining of human claudin-5 clusters in iBMECs showed only clustered patterns and density per cluster reached ~33,141 signals/um^2^ on average,* demonstrating antibody accessibility to epitopes in clusters with density ~2 times higher than these of FP-tagged overexpression constructs (eYFP).

Since the FP-tagged overexpression approach did not provide conclusive answers, we decided not to include this new data in the manuscript but to keep our original discussion element reflecting our limited conclusions:

“we cannot exclude that our staining approach underrepresents the entire claudin5 localizations (highly compacted foci might reduce antibodies accessibility)”.

If the reviewers and editor recommend including the data, we will be happy to do so.

A possible experimental setting that might answer this question would be to tag the endogenous protein in multiple locations and/or with linker-extensions and to test densities with antibody staining against the tags. This is an extension of the current study and would naturally require considerable time to produce the proper reagents and thus we feel should be included in a follow up study.

3. Temper conclusions regarding biology to better fit the presented data, especially with reference to overall BBB function since much of the data is developmental and observational. It's very important to reconcile the results shown with the conclusions being drawn especially in relation to the claudin-5 levels correlating with TJ functionality. The data shown in Figure 2 relate to developmental TJ functionality and this should be clarified and conclusions tempered. For Figure 4, all conclusions need to reference that these data relate to the developmental and early post-natal BBB which will be distinct from the adult BBB.

As suggested, we adjusted our conclusions with reference to overall BBB function to better emphasize that since much of the data is developmental; there should be further exploration of our findings in the adult and in different settings affecting barrier function in health and disease (lines 328-335).

We now provide more accurate conclusions, especially in relation to the claudin-5 levels correlating with TJ functionality (related to data in old Figure 2 – see lines, 184-186, also see lines, 328-335), and clearly indicate that our data relate to the developmental and early post-natal BBB, which might be distinct from the adult BBB (related to data in old Figure 4- lines 184-186 and 263-264).

Revisions Expected in follow up work:1. Provide in vitro co-localization with other junction proteins to begin to address mechanism. For example, how does the compacted localization of Claudin-5, ZO1 and Occludin relate to the other junctional structures such as the adherens junctions? If ZO-1 does not colocalize with Claudin-5 (or Occludin), then which receptor does it bind to that would explain the clustering?

We appreciate the suggestion which are all highly relevant and intriguing. As suggested, we will attempt to perform all of them and report as soon as we complete them.